# A randomized control trial of high-dose micronutrient-antioxidant supplementation in healthy persons with untreated HIV infection

Wendy L. Wobeser[1], Joanne E. McBane[2,3], Louise Balfour[2,4], Brian Conway[5], M. John Gill[2,6], Harold Huff[7], Donald L. P. Kilby[8], Dean A. Fergusson[9], Ranjeeta Mallick[10], Edward J. Mills[2,11], Katherine A. Muldoon[3,12,13,14], Anita Rachlis[2,14], Edward D. Ralph[15], Ron Rosenes[2], Joel Singer[2,16], Neera Singhal[11], Darrell Tan[2,17], Nancy Tremblay[3], Dong Vo[10], Sharon L. Walmsley[2,14], D. William Cameron[2,3,4,9]*, for the MAINTAIN Study Group

1 Department of Biomedical and Molecular Sciences and Public Health, Queen's University, Kingston, Ontario, Canada, 2 CIHR Canadian HIV Trials Network (CIHR-CTN), Vancouver, British Columbia, Canada, 3 Ottawa Hospital Research Institute, Ottawa, Ontario, Canada, 4 Division of Infectious Diseases, Department of Medicine, University of Ottawa at The Ottawa Hospital, Ottawa, Ontario, Canada, 5 Vancouver Infectious Disease Clinic, Vancouver, British Columbia, Canada, 6 Department of Microbiology, Immunology and Infectious Diseases, University of Calgary, Calgary, Alberta, Canada, 7 Canadian College of Naturopathic Medicine, Toronto, Ontario, Canada, 8 Faculty of Health Services, University of Ottawa, Ottawa, Ontario, Canada, 9 Clinical Epidemiology Program (CEP), University of Ottawa at The Ottawa Hospital Research Institute (OHRI), Ottawa, Ontario, Canada, 10 Ottawa Methods Centre, Clinical Epidemiology Program, Ottawa Hospital Research Institute, Ottawa, Ontario, Canada, 11 Global Evaluative Sciences, Vancouver, British Columbia, Canada, 12 Obstetrics and Maternal Investigations Research Group, The Ottawa Hospital, Ottawa, Canada, 13 School of Epidemiology, Public Health and Preventive Medicine, University of Ottawa, Ottawa, Canada, 14 Division of Infectious Diseases, Department of Medicine, University of Toronto, Toronto, Ontario, Canada, 15 Division of Infectious Diseases, Department of Medicine, University of Western Ontario, London, Ontario, Canada, 16 Faculty of Medicine, School of Population and Public Health, University of British Columbia, Vancouver, British Columbia, Canada, 17 La Ka Shing Knowledge Institute, St. Michael's, Toronto, Ontario, Canada

* bcameron@toh.ca

**Data Availability Statement:** All relevant data are within the paper and its Supporting information files.

## Abstract

### Background

Although micronutrient and antioxidant supplementation are widely used by persons with human immunodeficiency virus (HIV), a therapeutic role beyond recommended daily allowances (RDA) remains unproven. An oral high-dose micronutrient and antioxidant supplement (Treatment) was compared to an RDA supplement (Control) for time to progressive immunodeficiency or initiation of antiretroviral therapy (ART) in people living with HIV (PLWH).

### Methods

This study was a randomized, double-blind, placebo-controlled multicenter clinical trial. PLWH were recruited from Canadian HIV Trials Network sites, and followed quarterly for two years. Eligible participants were asymptomatic, antiretroviral treatment (ART)-naïve, HIV-seropositive adults with a CD4 T lymphocyte count (CD4 count) between 375–750 cells/μL. Participants were randomly allocated 1:1 to receive Treatment or Control

**Funding:** DWC and SW were supported by grant and salary awards of the Ontario HIV Treatment Network / Ontario Ministry of Health and DWC by a salary award from the Department of Medicine, University of Ottawa at The Ottawa Hospital. A CIHR postdoctoral fellowship award supported KAM. The funders had no role in study design, data collection and analysis, decision to publish, or preparation of the manuscript.

**Competing interests:** The work has been presented in part at the International AIDS Conference, Durban, South Africa, 22-27 July, 2017. The Data Safety Monitoring Board was struck at the National Centre of the CIHR-CTN, had full access to interim data at interim analysis, and conducted planned, independent unblinded analyses at the request of the PI (DWC). The trial sponsor (OHRI) and the investigators own the dataset,analyzed and retain the final data, and have sole responsibility and independent control over all analyses. There are no patents, products in development or marketed products to declare. This does not alter our adherence to PLOS ONE policies on sharing data and materials.

**Abbreviations:** AE, Adverse Event; AIDS, Acquired Immunodeficiency Syndrome; ART, Antiretroviral Therapy; BMI, Body Mass Index; CDC, Center for Disease Control; CI, Confidence intervals; CIHR, Canadian Institutes of Health Research; CTN, Canadian HIV Trials Network; DSMB, Data and Safety Monitoring Board; GCP, Good Clinical Practice; GTS, General Treatment Scale; HATS, HIV treatment adherence scale; HIV, Human Immunodeficiency Virus; HR, Hazard Ratio; IQR, Interquartile range; JAMA, Journal of the American Medical Association; LCI, Lower confidence interval; OHRI, Ottawa Hospital Research Institute; OHTN, Ontario HIV Treatment Network; QOL, Quality of life; RCT, Randomized controlled trial; RDA, Recommended daily allowances; RNA, Ribonucleic acid; SAE, Serious adverse event; SEM, Standard Error of the Mean; TCPS, Tri-Council Policy Statement; UCI, Upper confidence interval.

supplements. The primary outcome was a composite of time-to-first of confirmed CD4 count below 350 cells/µL, initiation of ART, AIDS-defining illness or death. Primary analysis was by intention-to-treat. Secondary outcomes included CD4 count trajectory from baseline to ART initiation or two years. A Data and Safety Monitoring Board reviewed the study for safety, recruitment and protocol adherence every six months.

## Results

Of 171 enrolled participants: 66 (38.6%) experienced a primary outcome: 27 reached a CD4 count below 350 cells/µL, and 57 started ART. There was no significant difference in time-to-first outcome between groups (Hazard Ratio = 1.05; 95%CI: 0.65, 1.70), or in time to any component outcome. Using intent-to-treat censoring, mean annualized rates of CD4 count decline were -42.703 cells/µL and -79.763 cells/µL for Treatment and Control groups, with no statistical difference in the mean change between groups (-37.06 cells/µL/52 weeks, 95%CI: (-93.59, 19.47); p = 0.1993). Accrual was stopped at 171 of the 212 intended participants after an interim analysis for futility, although participant follow-up was completed.

## Conclusions

In ART-naïve PLWH, high-dose antioxidant, micronutrient supplementation compared to RDA supplementation had no significant effect on disease progression or ART initiation.

## Clinical trial registration

**ClinicalTrials.gov Identifier**: NCT00798772.

## Introduction

Clinically, micronutrient deficiencies have been correlated with faster disease progression, more frequent opportunistic infections, and a higher incidence of human immunodeficiency virus (HIV)-attributable mortality [1,2]. Low serum micronutrient concentrations and high oxidative stress in HIV infection have been associated with impaired immunity, CD4 T lymphocyte depletion, and inflammation [3]. HIV infection triggers oxidative stress in the body through increased production of reactive oxygen species and dysregulation of antioxidant responses [4]. HIV-infected individuals demonstrate reduced antioxidant capacity and decreased serum levels of important antioxidants such as glutathione and selenium [4,5].

Some beneficial effects of micronutrient and antioxidant supplementation on HIV-associated outcomes have been documented, primarily in low-income countries with high HIV prevalence. Evidence favoring targeted micronutrient and antioxidant supplementation in improving survival and slowing disease progression is promising for children [6,7] and pregnant women [8] with HIV infection, and for individuals with lower micronutrient levels [9], or advanced HIV and CD4 T lymphocyte counts below 200 cells/µL on antiretroviral therapy (ART) [10]. Findings from a randomized controlled trial (RCT) in Botswana revealed that ART-naïve HIV patients receiving 24 months of micronutrient supplementation had a significantly lower risk of reaching CD4 T lymphocyte counts ≤250 cells/µL, and experiencing AIDS-related conditions or death, when compared to a placebo group [11]. An American study investigating the impact of a 12-week course of broad-spectrum micronutrient

supplementation in HIV-patients on ART showed a significant 24% increase in baseline CD4 T lymphocyte count in the micronutrient group versus 0% change in the placebo group [2].

An RCT among ART-naïve adults with malnutrition in Tanzania and Zambia demonstrated that a lipid-based nutritional supplement with additional vitamins and minerals significantly increased mean CD4 T lymphocyte count at 12 weeks post-ART compared to a lipid-based nutritional supplement alone, however high-dose supplements did not significantly improve clinical outcomes including mortality and CD4 T lymphocyte count on ART [12]. An RCT investigating people living with HIV in Uganda did not demonstrate an impact of multivitamin supplementation on the rate of CD4 T lymphocyte recovery in a group of persons recently started on ART [13]. Similarly, an RCT in Rwanda found that 24 months of selenium supplementation had no significant effect on CD4 T lymphocyte depletion to <350 cells/μL, ART initiation, emergence of a documented CDC AIDS-defining illness, or viral suppression in HIV-infected patients [14].

The primary objective of the MAINTAIN study (CTN238) [15,16] was to assess the efficacy of a high-dose antioxidant and micronutrient supplement compared to standard recommended daily allowance (RDA) multivitamins in slowing disease progression or immune deficiency among people living with HIV, in a rigorous RCT in Ottawa, Canada.

## Methods

### Study design

This was a randomized, double-blind, multicenter controlled clinical trial. A total of 171 consenting, eligible, ART-naïve, asymptomatic HIV-positive adults were enrolled and randomly allocated (1:1) to receive a high-dose micronutrient-antioxidant preparation (K-PAX Ultra®, Treatment) or 100% RDA multivitamins (identically appearing and packaged, Control). The MAINTAIN study protocol is registered and published in detail and includes a complete description of the study medication formulations [15] and key micronutrient levels are listed in Table 1. Study supplements were subjected to regular stability analyses reported from a laboratory (American Analytical Chemistry Laboratories Corp.) independent of the manufacturers and the investigators. S1 Table has the summary of the treatment stability reports.

The MAINTAIN study randomization process used a web-based system. The Ottawa Methods Centre statistician, not associated with the study, used SAS® software to generate the randomization allocation codes. The randomization schema was then loaded into a SQL database table/server. The web-based randomization was tested thoroughly and validated before it was put in use. The MAINTAIN study web-based randomization site is secured with Entrust SSL 256-bit encryption. It was fully protected virtually and physically. The MAINTAIN randomization website was accessed by the study coordinators when a participant was deemed eligible for the study. No participant personal health information was collected.

**Table 1. Micronutrient dose per 16 capsules (actual daily dose) of indicated micronutrients in control or treatment supplements.**

| | Vitamin B$_6$ (mg) | Carotenoids (IU) | Vitamin C (mg) | Iron (mg) | Selenium (μg) | Zinc (mg) | Acetyl-L-Carnitine (mg) | Lipoic Acid (mg) | n-Acetyl Cysteine (mg) |
|---|---|---|---|---|---|---|---|---|---|
| Control [1,2] | 2 | 3,500 | 60 | 18 | 20 | 15 | 0 | 0 | 0 |
| Treatment [1,3] | 200 | 20,000 | 2,000 | 18 | 200 | 30 | 1,000 | 400 | 1,200 |

[1]The readings are given in different units based on the micronutrient per 16 capsules (daily dose of 8 capsules taken twice a day for 96 weeks). Original amount of each micronutrient as reported by the manufacturer's label for the daily dose of 16 capsules (15).

[2]Control refers to the 100% recommended daily allowance supplement.

[3]Treatment refers to the high-dose antioxidant-micronutrient supplement (K-PAX Ultra®).

Allocation was conducted using a permutated blocked central randomization method (blocks of size four) stratified by centre, and by CD4 T lymphocyte count above and below 500 cells/μL. Participants were recruited from 21 separate CIHR-CTN supported clinical sites across Canada. Participants were approached during routine clinic visits and consenting participants were screened. Eligible participants were randomized to 1:1 to receive the Treatment or Control supplements and were followed-up on protocol during regular quarterly clinic visits (approximately every 12 weeks) for two years (during the period from March 2009 till June 2013). Inclusion and exclusion criteria are already described previously [16] (PLOS ONE). In brief -Volunteers were eligible to participate if they were asymptomatic HIV-infected individuals with a CD4 T cell count between 375–750 cells/mL at screening evaluation, and had never been on ART (excluding less than seven days duration, and perinatal transmission prophylaxis). Exclusion criteria were HIV-2 infection alone, pregnancy or not willing to practice barrier method of birth control, on treatment for ongoing opportunistic infection and taking micronutrient or natural health product supplements that may overlap study medication components within thirty days of randomization.

## Measures

The primary composite outcome was time from randomisation to a composite of the first of any one of the following four primary outcome measures: CD4 T lymphocyte count >350 cells/μL (confirmed by a second measure one month following the first low measure), start of ART as a clinical decision of the treating physician and the patient, emergence of documented CDC-defined AIDS-defining illness, or death.

Demographic information collected included age in years at baseline (as median, interquartile range (IQR)), sex (male/female), and ethnicity (Asian, Black, White, Indigenous, Other). The following continuous clinical variables were collected at baseline and each visit (every 12 weeks) up to 96 weeks: Body Mass Index (BMI), HIV-1 RNA (copies/mL), CD4 T lymphocyte count (cells/μL), CD8 T lymphocyte count (cells/μL) and CD4:CD8 (T lymphocyte ratio).

The following serum micronutrient levels were measured: carotene (μmol/L), hydroxy-vitamin D (nmol/L), vitamin $B_{12}$ (pmol/L) and folate (serum nmol/L or erythrocyte ng/L). Low levels of micronutrients were defined at the following thresholds: carotene <1 mmol/L, vitamin D <75 nmol/L (insufficient), <40 nmol/L (deficient) or <20 nmol/L (significantly deficient) of 25-OH vitamin D, and vitamin $B_{12}$ < 133 pmol/L. Folate thresholds were either <15 nmol/L or <450 ng/L depending on the serum or erythrocyte assay used.

Several secondary outcome measures were collected over the course of the study including: CD4 T lymphocyte count, CD8 T lymphocyte count, CD4:CD8 ratio, HIV viral load, serum chemistries and quality of life (QOL). These secondary outcomes were measured at each study visit, occurring every 12 weeks for up to 96 weeks. Measurements were taken even after a primary outcome or the participant was off-protocol; however reported data are censored either: 1) for intent-to-treat (start ART censored), or 2) for those off-protocol (those with a protocol violation or primary outcome censored). There was a total of 171 participants in the study with 81 participants allocated to the Control group and 90 participants allocated to the Treatment group; however participant number decreased as those with primary outcomes or off-protocol grew.

The serum chemistries measured include: albumin (g/L); alanine aminotransferase (ALT, international units/L–IU/L); alkaline phosphatase (IU/L); amylase (IU/L); aspartate aminotransferase (AST, IU/L); bilirubin (total, μmol/L); blood glucose, random (mmol/L); blood urea nitrogen (BUN, mmol/L); C-reactive protein (mg/L); creatinine (mmol/L); and total protein (g/L).

The EuroQoL group three level survey was used to survey participants for five parameters associated with QOL (EQ-5D-3L); activity, anxiety, mobility, pain and self-care. Each participant rated their QOL with regards to the parameter as: 1) no problem; 2) some problems; or 3) extreme problems. All QOL measurements were taken at each study visit (every 12 weeks) until 96 weeks or participant discontinuation. For simplicity, the categories of 2) some problems and 3) extreme problems were combined.

Study supplement adherence was calculated using the residual pill count at each return visit. Self-reported study supplement adherence was evaluated using the HIV Treatment Adherence Scale (HATS) [17] assessing missed doses in the previous weeks and the General Treatment Scale (GTS) [18], a validated questionnaire measuring study drug or supplement adherence. GTS scores range from 0 (low adherence) to 25 (high adherence) and are calculated by summing a total of 5 questions on a 5-point Likert scale. In June 2013, a pre-planned mid-study sub-analysis was conducted among 127 participants in the MAINTAIN study which confirmed sufficiently high levels of adherence (88%), which with pharmaceutical stability of the formulation might allow a meaningful comparison of the interventions on the outcomes in the study [16].

There have been a total of 21 serious adverse events reported, related to 15 individual participants, none were deemed to be treatment related.

## Statistical analyses

Sample size was calculated as previously described [15,16]. From a previous ART-naïve HIV-positive cohort in the UK it was estimated that 45% of the study population would experience at least one primary outcome (CD4 T lymphocyte >350 cells/µL or start ART) by 2 years [19]. A two-sided log rank test calculated that a total sample size of 218 participants (109 participants per group) achieves 80 percent power at a 0.05 significance level to detect a difference of 18% between the proportions surviving (no primary outcome) in the respective groups after 2 years. An accrual time of one year with 50% enrolment by six months was assumed, and a modest non-compliance rate of 15% for the Treatment group and 20% for the Control group was employed based on previous trials [9,15,19].

Statistical analyses of primary outcomes were conducted on an intention-to-treat basis with all participants analyzed according to the intervention to which they were allocated, regardless of adherence. Survival analysis was used to analyze primary outcomes. The between group difference in primary outcome, time to the first of the composite which included confirmed CD4 T lymphocyte count below 350 cells/µL, emergence of CDC-defined AIDS-defining illness, start of ART or death, was analyzed using a two-sided log-rank test. Participants lost to follow-up prior to reaching a primary outcome were censored at that point.

Time to first event was established and compared using a two-sided log-rank test, and was conducted with the composite outcome primarily, and secondarily for each individual outcome. Cox proportional hazard models were conducted to estimate effect size. Hazard Ratios (HR) and 95% confidence intervals (CI) are presented. Secondary sensitivity analyses were conducted censoring participants at the point of protocol violations and non-adherence (censoring for those off-protocol).

For CD4% T lymphocyte analysis and all secondary outcomes analyses (excluding QOL), linear mixed-effects models were used to analyze censored participants data including random effects models to account for repeated measured correlation at the patient level for these analyses. Data for CD4% was censored for intent-to-treat only. CD4 trajectory and HIV viral load were censored for both intent-to-treat and off-protocol. Notably only those with measurements out to 36 weeks were included in those analyses. The CD8, CD4:CD8 ratio, and serum

chemistry data was censored for those off-protocol. The mean slopes after data censoring were reported over 52 weeks for Control and Treatment groups. The differences between mean slopes for Control versus Treatment groups are reported with the 95% confidence interval being given for each as well as the p-value. An alpha level of 0.05 was considered significant.

QOL data was censored for measurements taken when the participant was off-protocol and reported as a ratio of those with no problems compared to the total persons reporting. No statistics were performed on QOL data.

All analyses were performed using SAS® version 9.3 software.

### Institutional review

This study was conducted in accordance with Health Canada's Good Clinical Practice (GCP) guidelines [20], Health Canada Food & Drug Regulations, Part C Division 5 [21], and in accordance with the Declaration of Helsinki and the Tri-Council Policy Statement: Ethical Conduct for Research Involving Humans (TCPS) [22]. The research ethics board of each participating site approved the study protocol and informed consent forms. Written informed consent was obtained from each participant.

## Results

We enrolled 171 participants between March 2009–2011, 90 in the Treatment group and 81 in the Control group. The administered supplements in both study arms remained stable between batches and across the study (S1 Table). The CONSORT diagram (Fig 1) includes details of study enrolment and participation. Overall, there were 95 (55.6%) participants who completed the study to primary outcome or 96 weeks, with 33 supplement discontinuations in the Treatment group and 26 in the Control group. Multiple reasons could be indicated and the most common reasons for premature discontinuation included: withdrawal of consent to study supplement [33], adverse events [11], and protocol non-compliance [6]. Of the 59 participants who discontinued prematurely, 46 participants had documented protocol violations (Treatment group-29; Control group 17) including: loss to follow-up, missing visits, and non-adherence. Using residual pill count at each return visit, study supplement adherence was estimated to be 85.3%. Adherence scale metrics and imputed estimates have been previously reported from this trial [16], and our final adherence levels were similar.

### Interim analyses

An interim analysis was conducted to assess safety, clinical efficacy and futility. This was partially prompted by slowed recruitment that arose during the trial. The analysis was run on 127 participants with complete data, by an independent blinded statistician and evaluated by the Data Safety and Monitoring Board (DSMB).

Two sets of simulations were carried out to evaluate the probability of detecting a significant clinical effect of the high-dose supplement. The first scenario assumed no further recruitment and followed the remaining participants until the end of year 2 or reaching an outcome. The second scenario assumed that the trial continues to recruit until the enrolment target was reached and all participants were followed until the end of year 2 or reaching an outcome. The results indicated no significant difference between the Control and the Treatment groups. The probability of detecting a statistically significant difference by the end of the study was less than 0.01 in both scenarios.

The DSMB for the MAINTAIN study reviewed the results and recommended that the accrual be halted early for futility as there was no evidence of a treatment effect, although completion of follow-up was recommended.

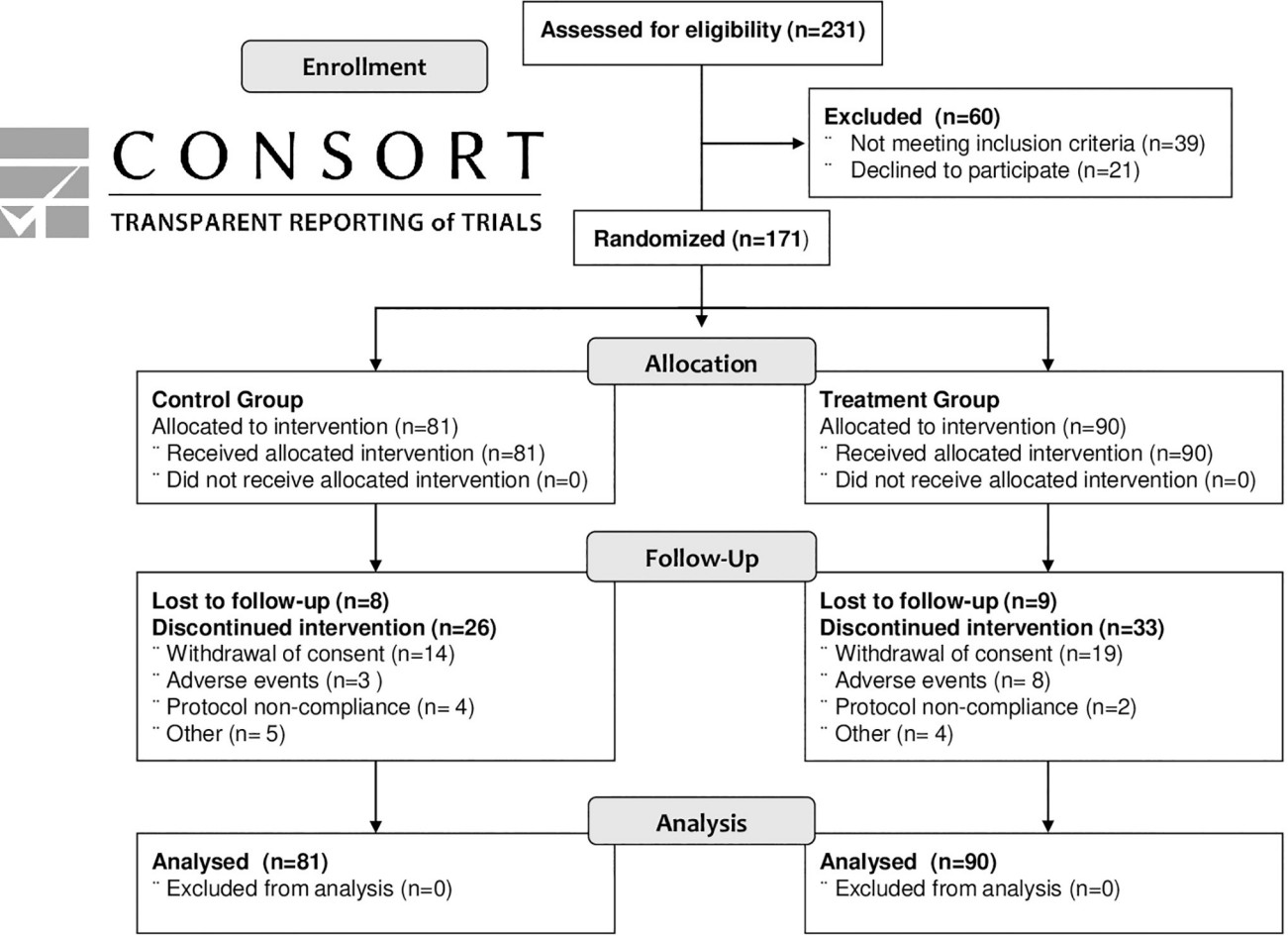

**Fig 1. CONSORT flow diagram.** Enrollment, eligibility screening, allocation, follow-up, and analysis of participants in the MAINTAIN Randomized Controlled Trial.

## Demographic characteristics

Table 2 displays the demographic and clinical characteristics of the 171 participants enrolled in the study. Overall, the study population median age was 37.9 years (IQR: 30.2–45.9), 83.0% male, and predominantly White (70.2%), followed by Black (15.2%) and Asian (3.5%). At baseline, the median CD4 T lymphocyte count was 495 cells/μL (IQR: 429–610), and plasma viral load was 11057 copies/mL (IQR: 2894–29746).

## Micronutrient levels

Table 3 displays the micronutrient levels at baseline for carotene (μmol/L), vitamin D (nmol/L), vitamin $B_{12}$ (pmol/L), and folate (nmol/L or ng/L). Low carotene levels below 1 mmol/L were found in 22.9% in the Treatment arm and 23.3% in the Control arm at baseline. There was a downward trend from baseline in prevalence of low carotene levels in the Treatment group at year 1 and both groups in year 2. At baseline, a large proportion of enrolled persons had low levels of vitamin D (<75 nmol/L), 67.5% and 69.0% for the Treatment and Control groups respectively. Vitamin $B_{12}$ deficiency was below 5% in both groups at all time points.

**Table 2. Demographic and clinical characteristics for randomized participants at baseline (n = 171).**

| Demographic Variables | Total | Control | Treatment |
|---|---|---|---|
| **Age (Median (IQR))** | 37.9 (30.2–45.9) | 37.4 (29.9–43.3) | 38.4 (30.3–46.3) |
| **Sex (n(%))** | | | |
| Male | 142 (83.0) | 64 (79.0) | 78 (86.7) |
| Female | 29 (17.0) | 17 (21.0) | 12 (13.3) |
| **Ethnicity (n(%))** | | | |
| Asian | 6 (3.5) | 3 (3.7) | 3 (3.3) |
| Black | 26 (15.2) | 10 (12.4) | 16 (17.8) |
| White | 120 (70.2) | 61 (75.3) | 59 (65.6) |
| Other | 19 (11.1) | 7 (8.6) | 12 (13.3) |
| **Clinical Characteristics** | **Median (IQR)** | **Median (IQR)** | **Median (IQR)** |
| BMI (kg/m$^2$) | 25.7 (22.9–28.4) | 26.1 (22.4–28.4) | 25.3 (23.0–28.6) |
| HIV-1 RNA (copies/mL) | 11057 (2894–29746) | 12390 (2804–26896) | 12390 (3432–31504) |
| CD4 T lymphocytes (cells/μL) | 495 (429–610) | 500 (429–654) | 493.5 (430–580) |
| CD8 T lymphocytes (cells/μL) | 879 (648–1156) | 905 (638–1229) | 856 (656–1080) |
| CD4:CD8 (T lymphocyte ratio) | 0.60 (0.42, 0.80) | 0.60 (0.40 0.81) | 0.62 (0.43, 0.80) |

Serum folate levels below 15 nmol/L were seen in 14.7% and 15.1% Treatment and Control groups respectively at baseline, which was stable in prevalence in follow-up.

## Safety and adverse events

There was a total of 29 serious adverse events (SAEs) reported related to 19 (11.1%) individual participants; none were deemed to be supplement-related by the DSMB. SAEs included appendicitis, community-acquired pneumonia, cholecystitis, perianal abscess, attempted suicide, fever, lymphadenopathy, benzodiazepine overdose, depression, septic arthritis, intra-abdominal abscess, and psychosis. There were no deaths. See S2 Table for the complete list of all 29 SAEs. There were two cases of lymphoma or suspected lymphoma in the Treatment group, but both were off-protocol before the event (one for early and complete non-adherence, and one on ART for prior primary outcome), and therefore neither was a primary outcome. In one case, the participant had started ART after 84 weeks and was diagnosed with biopsy-proven Stage IIA Non-Hodgkins lymphoma at the 96-week visit, about 661 days after the first dose. In the other case, the participant withdrew prior to the 12-week visit due to pill burden. The off-protocol participant presented with a suspected lymphoma (fever and lymphadenopathy) 91 days after randomization and was started on ART (Atripla). In this case the participant was subsequently lost to follow-up and the diagnosis was not reported by biopsy.

## Primary and secondary outcomes

A total of 66 individuals had at least one component of primary study outcome: 27 individuals reached a CD4 T lymphocyte count below 350 cells/μL, 57 individuals started ART, and no individual reached an AIDS-defining event as a primary outcome, and no one died during the study period. There were no significant differences in time to first outcome identified in survivorship analysis. Fig 2 displays time to CD4 T lymphocyte decline below 350 cells/μL: log-rank test p = 0.3033; HR = 1.48 (95% CI: 0.69, 3.20), p = 0.3142. Fig 3 displays time to ART initiation regardless of CD4 T lymphocyte count: log-rank test p = 0.7943; HR = 0.94 (95% CI: 0.56, 1.57), p = 0.8013. Fig 4 displays the primary analysis of time to first of any event: log-rank test p = 0.8387; HR = 1.05 (95% CI: 0.65, 1.70), p = 0.8452.

**Table 3. Number of MAINTAIN participants with micronutrient deficiencies over the study period (n = 171)[1].**

| | Baseline Levels | | |
| --- | --- | --- | --- |
| | **Total** | **Control** | **Treatment** |
| **Carotene < 1µmol/L** | 36/156 (23.1%) | 17/73 (23.3%) | 19/83 (22.9%) |
| **Vitamin D nmol/L–mean (std)** | 64.8 (28.1) | 64.9 (26.1) | 64.8 (29.9) |
| **Vitamin D < 75 nmol/L** | 105/154 (68.2%) | 49/71 (69.0%) | 56/83 (67.5%) |
| **Vitamin D <40 nmol/L** | 33/154 (21.4%) | 14/71 (19.7%) | 19/83 (22.9%) |
| **Vitamin D <20 nmol/L** | 4/154 (2.6%) | 0/71 (0%) | 4/83 (4.8%) |
| **Vitamin B12<133 pmol/L** | 6/164 (3.7%) | 3/76 (4.0%) | 3/88 (3.4%) |
| **Folate < 15 nmol/L (serum)** | 18/121 (14.9%) | 8/53 (15.1%) | 10/68 (14.7%) |
| **Folate < 450 ng/mL (erythrocyte)** | 3/15 (20.0%) | 1/7 (14.3%) | 2/8 (25.0%) |
| **Folate < 15 nmol/L or 450 ng/mL[2]** | 21/136 (15.4%) | 9/60 (15.0%) | 12/76 (15.8%) |
| | Trial Year 1 | | |
| | **Total** | **Control** | **Treatment** |
| **Carotene < 1µmol/L** | 17/111 (15.3%) | 13/56 (23.3%) | 4/55 (7.3%) |
| **Vitamin D nmol/L–mean (std)** | 82.1 (29.3) | 83.7 (27.6) | 80.4 (31.2) |
| **Vitamin D < 75 nmol/L** | 39/96 (6.2%) | 21/49 (42.9%) | 18/47 (38.3%) |
| **Vitamin D <40 nmol/L** | 6/96 (1.0%) | 0/49 (0.0%) | 6/47 (12.8%) |
| **Vitamin D <20 nmol/L** | 1/96 (1.0%) | 0/49 (0.0%) | 1/47 (2.1%) |
| **Vitamin B12<133 pmol/L** | 3/113 (2.6%) | 3/58 (5.2%) | 0/55 (0.0%) |
| **Folate < 15 nmol/L (serum)** | 9/76 (11.8%) | 5/37 (13.5%) | 4/39 (10.3%) |
| **Folate < 450 ng/mL (erythrocyte)** | 0/15 (0.0%) | 0.9 (0.0%) | 0.6 (0.0%) |
| **Folate < 15 nmol/L or 450 ng/mL[2]** | 9/91 (9.9%) | 5/46 (10.9%) | 4/45 (8.9%) |
| | Trial Year 2 | | |
| | **Total** | **Control** | **Treatment** |
| **Carotene < 1µmol/L** | 14/120 (11.7%) | 8/58 (13.8%) | 6/62 (9.7%) |
| **Vitamin D nmol/L–mean (std)** | 150.6 (617.4) | 202.5 (861.5) | 98.7 (149.2) |
| **Vitamin D < 75 nmol/L** | 48/102 (47.1%) | 22/51 (43.1%) | 26/51 (51%) |
| **Vitamin D <40 nmol/L** | 5/102 (4.9%) | 4/51 (7.8%) | 1/51 (1.2%) |
| **Vitamin D <20 nmol/L** | 0/102 (0%) | 0/51 (0%) | 0/51 (0.0%) |
| **Vitamin B12<133 pmol/L** | 0/123 (0%) | 0/61 (0%) | 0/62 (0.0%) |
| **Folate < 15 nmol/L (serum)** | 14/81 (17.3%) | 6/38 (15.8%) | 8/43 (18.6%) |
| **Folate < 450 ng/mL (erythrocyte)** | 0/12 (0.0%) | 0/88 (0.0%) | 0/4 (0.0%) |
| **Folate < 15 nmol/L or 450 ng/mL[2]** | 14/93 (15.1%) | 6/46 (13.0%) | 8/47 (17.0%) |

[1]The number of participants measured for each micronutrient are listed, but the total number of participants recruited who were eligible after screening and were allocated to Control (n = 81) or Treatment (n = 90) groups was 171.

[2]Folate levels were measured differently across sites and combined in the final variable.

In sensitivity analyses where participants were censored if they went off-protocol, no significant differences between study arms were detected for CD4 T lymphocyte count below 350 cells/µL (p = 0.2451, S1 Fig), ART initiation (p = 0.9936, S2 Fig) or any event (p = 0.7059, S3 Fig).

## Secondary outcome measures

Secondary outcome measures include the rate of change or trajectory of CD4 T lymphocytes, CD8+ T lymphocytes, CD4:CD8 ratio, HIV viral load, serum chemistry and quality of life (QOL) surveys. Measurements were recorded at each quarterly study visit (12 weeks) until the end of study (discontinuation by the participant or 96 weeks) for Control (n = 81) and

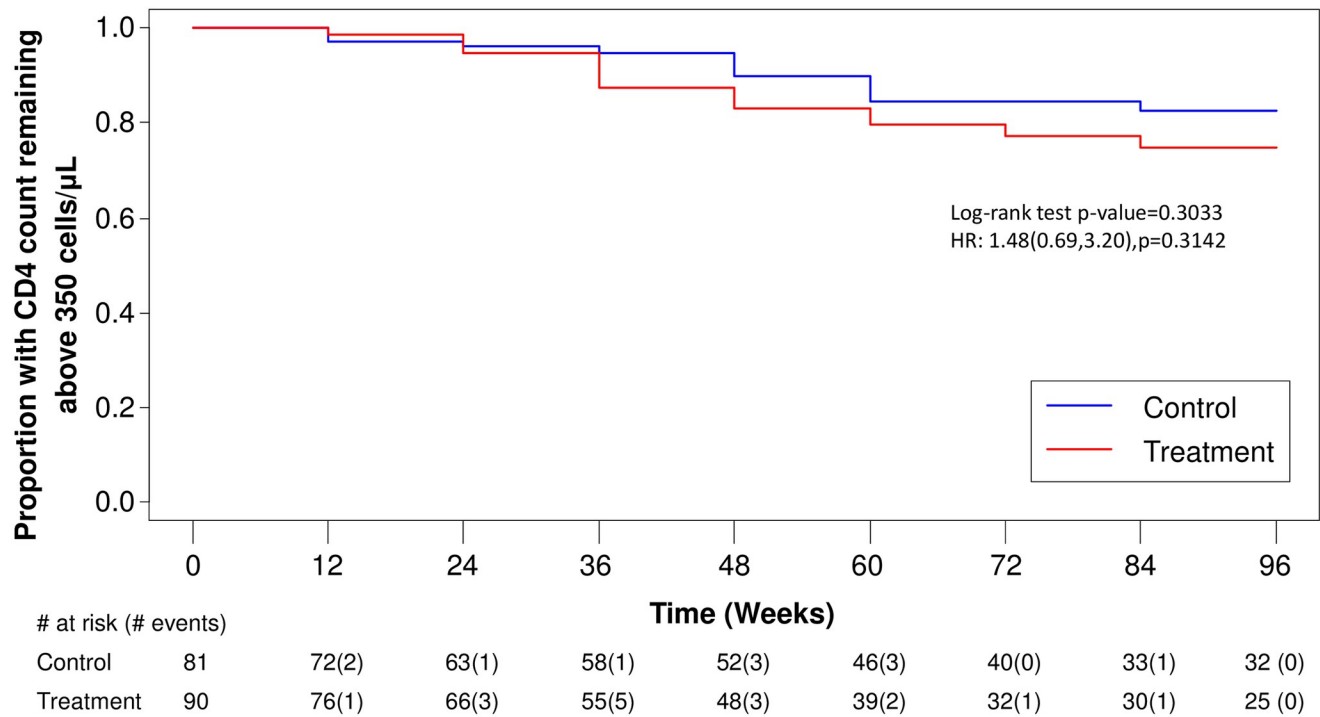

**Fig 2. Kaplan-Meier Curve: Time to confirmed**\* (\*in 2 measures) CD4 T lymphocyte count decline below 350 cells/μL. Week 0 includes all individuals who were screened in and allocated to a group. The table beneath shows the individuals remaining at risk at each time point and then number of individuals experiencing an event in that interval is in brackets. There were 11 events in the Control and 16 events in the Treatment group over the study period of 96 weeks.

Treatment (n = 90) groups. S4 Fig shows serial CD4 T lymphocyte count of data from those participants with measurements for at least 36 weeks (i.e. having at least 3 data points) censored only for participants who started ART (intent-to-treat) for Control (S4A Fig) and Treatment (S4B Fig) groups. We used linear mixed-effects models to analyze the CD4 T lymphocyte data and calculate the mean change over time for Control and Treatment participants identified in S4A–S4D Fig that were censored for intent-to-treat (ITT; S4A and S4B Fig) or per-protocol analysis (OP; S4C and S4D Fig). S4E Fig shows the mean change in CD4 T lymphocyte counts over time (cells per μL per 52 weeks) for ITT censoring was -79.7 cells/μL in the Control group and -42.7 cells/μL in the Treatment group and for OP analysis was -71.2 cells/μL in the Control group and -44.5 cells/μL in the Treatment group. The difference between the mean annualized rates was -37.1 (-93.6, 19.5) cells/μL (ITT) and -26.7 (-88.6, 35.1) cells/μL, but neither difference was statistically different (ITT p = 0.1993; OP 0.3976) S4F Fig.

The CD4 T lymphocyte % (of total CD3+ lymphocytes) declined at a mean rate of 0.53% and 0.31% every 12 weeks for the Treatment and Control groups respectively. The difference in decline was 0.22% (95% CI: 0.49,0.93) however the decline was not statistically different between the two allocations (p = 0.535).

CD8 T lymphocyte counts, CD4:CD8 T lymphocyte ratio data was censored for those participants OP.

Mean change in CD8 T lymphocyte counts over time was 2.5 cells/μL/52 weeks (Control) and -8.8 cells/μL/52 weeks (Treatment). S5A Fig shows the difference in mean slope between Control and Treatment groups was not statistically significant (p = 9187). The mean change in CD4:CD8 T lymphocyte count ratios over time was -0.072 ratio/52 weeks (Control) and -0.063

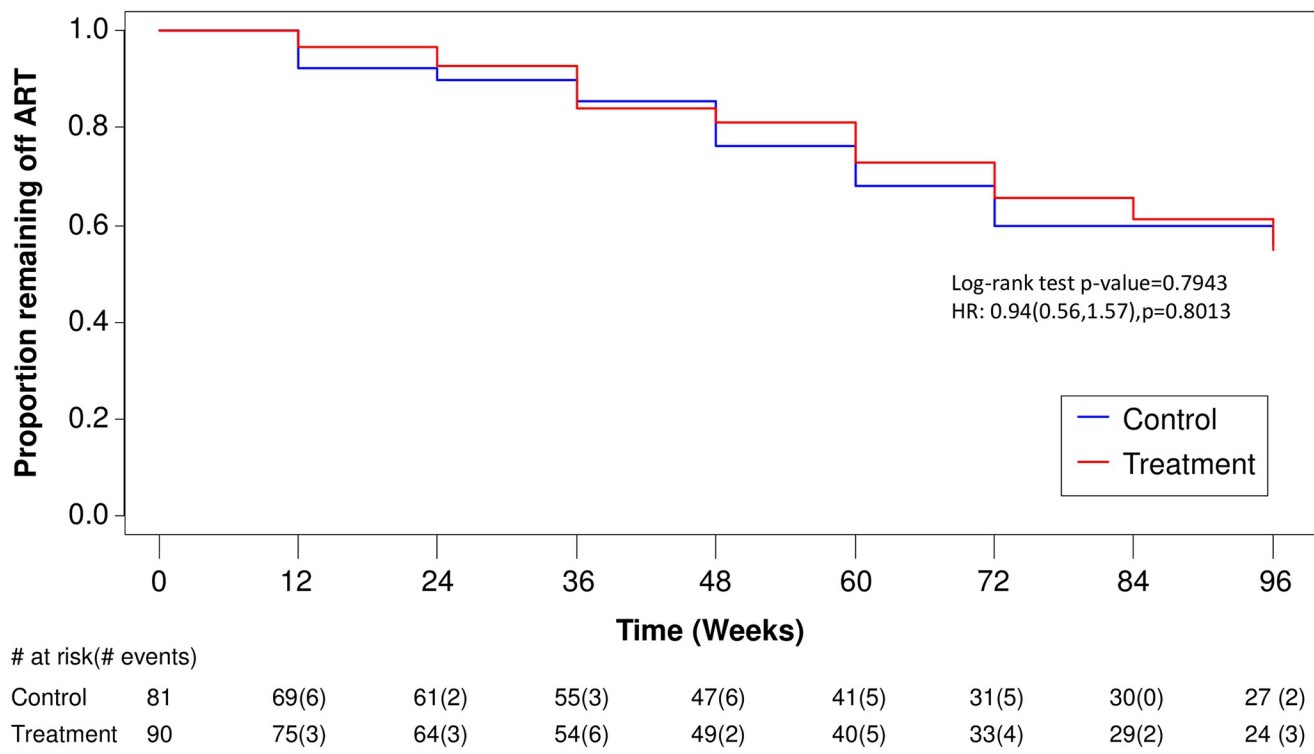

**Fig 3. Kaplan-Meier Curve: Time to start of ART regardless of CD4 T lymphocyte count.** Week 0 includes all individuals who were screened in and allocated to a group. The table beneath shows the individuals remaining at risk at each time point and the number of individuals remaining at risk at each time point and the number of individuals experiencing an event in that interval is in brackets. There were 29 events in the Control group and 28 events in the Treatment group over the study period of 96 weeks.

ratio/52 weeks (Treatment). S5B Fig shows the difference in mean slope between Control and Treatment groups was not statistically significant (p = 8615). This agrees with the CD4 analyses showing no significant effect of treatment in the ITT (Fig 2 and S4 Fig) and OP analyses (S1 and S4 Figs).

S6 Fig shows HIV viral load measurements taken for participants in Control and Treatment groups censored for ITT (S6A and S6B Fig) or OP (S6C and S6D Fig). Data was censored for those participants starting ART and those completing less than 36 weeks of measurements. S6E Fig shows the mean change in HIV viral load per annum (52 weeks) for participants depicted in S6A–S6D Fig. The mean annualized change in HIV viral load was 0.4 $Log_{10}$ copies/mL (Control group) versus 0.57 $Log_{10}$ copies/mL (Treatment group) with ITT censored data. The mean annualized change in HIV viral load was 0.31 $Log_{10}$ copies/mL (Control group) versus 0.43 $Log_{10}$ copies/mL (Treatment group) with OP censored data. The difference in mean annualized change (Control versus Treatment, S6F Fig) was not statistically significant in the ITT (p = 0.612) or the OP analysis (p = 0.7507).

Serum chemistries measured in S7 Fig include: albumin, ALT, alkaline phosphatase, amylase, AST, bilirubin (total), blood glucose (random), BUN, C-reactive protein, creatinine, and total protein. The mean change in serum chemistry per 52 weeks is reported in a table (S7A Fig) and the difference in mean change over 52 weeks is represented graphically. Although some serum chemistry measures increase with time, others decrease with time, there were no statistically significant differences between the mean rates of change for Control versus Treatment groups (S7 Fig).

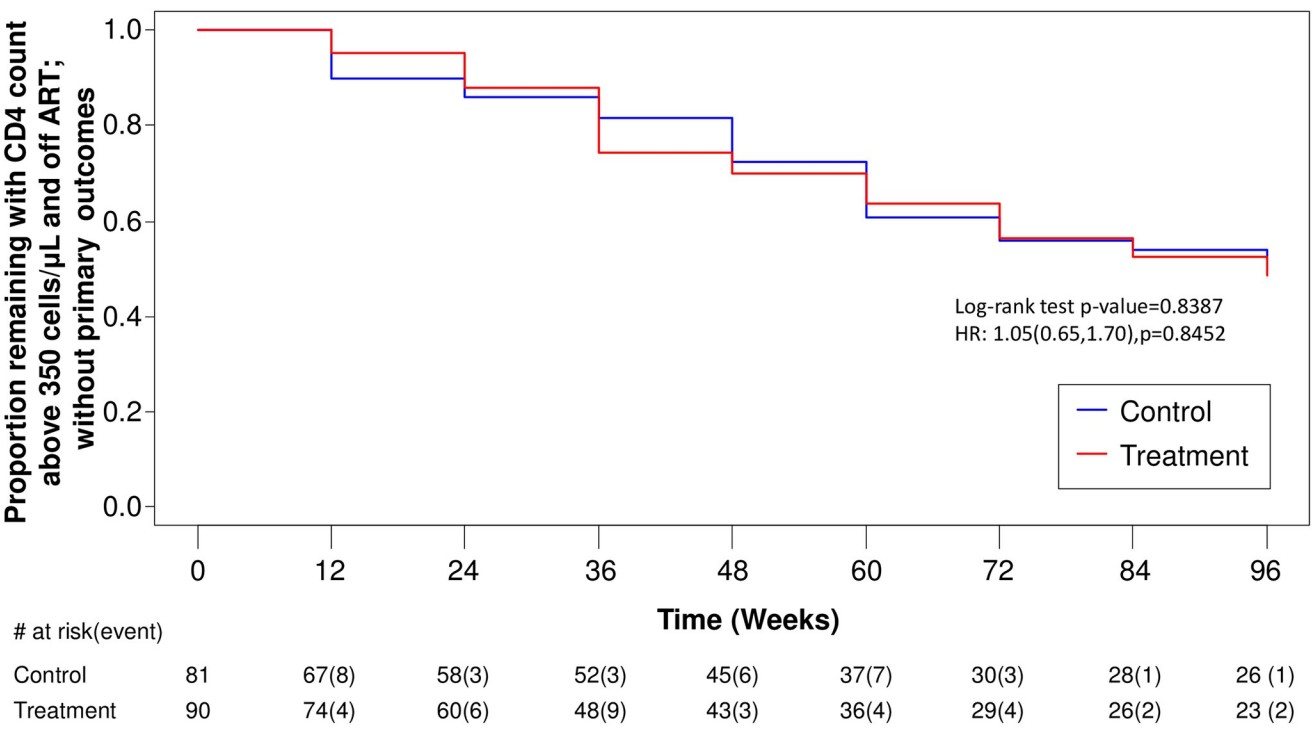

**Fig 4. Kaplan-Meier Curve: Time to any primary outcome event.** Week 0 includes all individuals who were screened in allocated to a group (CD4<350, ART, AIDS or death). The table beneath shows the individuals remaining at risk at each time point and the number of individuals experiencing an event at that time point is in brackets. There were 32 events in the Control group and 34 events in the Treatment group over the study period of 96 weeks.

To assess the participant's QOL, the participants were asked to rate five parameters associated with QOL including: activity (A), anxiety (B), mobility (C), pain (D) and self-care (E) (S8 Fig). These aspects were rated at each quarterly study visit as no problem, some problems or extreme problems (until discontinuation by participant or end of study 96-week visit) as per EuroQoL group three level survey (EQ-5D-3L). In S10 Fig, each parameter is graphed separately, with % of participants having either some or extreme problems (combined) with the parameter being plotted for Control versus Treatment groups as assessed at each study visit. There were no significant differences between groups for QOL parameters (S8 Fig). The rate of change of Control and Treatment groups of the QOL parameters were not significantly different from zero, suggesting that there was no change in QOL with time on either supplement (S8 Fig).

## Discussion

In this comparison of a standard RDA multivitamin to a high-dose multivitamin with antioxidants, we did not observe a significant change in HIV-disease progression as measured by CD4 T lymphocyte decline below 350 cells/μL, emergence of documented AIDS-defining illness, start of ART, or death. This rigorous trial and analysis contribute high quality data to a divided literature regarding the efficacy of multivitamins and antioxidants as supplemental treatment for people living with HIV in this North American setting. Our findings are congruent with a 2017 Cochrane review of RCTs in HIV positive adults (excluding pregnancies and those with metabolic morbidity related to ART) showing that no RCT to date has shown significant benefit of high-dose single, dual or multivitamin supplement in terms of preventing

CD4 T lymphocyte decline or preventing increase in HIV viral load versus standard supplement, although all evidence was reported with low certainty due to low power trials [23].

Use of supplements in the non-institutionalized, general American population stabilized between 1999 and 2012 with just over half of the population reporting some form of supplement use [24], resulting in a $32 billion industry with little evidence of benefit, and a slow regulatory response to harm when demonstrated [25]. Several rigorous trials and systematic reviews have shown that vitamin supplementation among adults without nutritional deficiencies did not reduce all-cause mortality, cardiovascular disease or cancer [26], and has no long-term effect on cognitive performance [27], and that high-dose multivitamins had no effect on cardiovascular disease compared to placebo [28]. Published studies have indicated that there is no substantial health benefit associated with multivitamin use and recommend against routine supplementation without indication of a specific micronutrient/antioxidants deficiency [27]. Evidence-based guidelines for women in pregnancy support the use of folic acid (to reduce the risk of fetal neural tube defect) and vitamin D supplementation for bone health, however there is insufficient evidence to show a clinical benefit for other supplements in well-nourished women [29]. Some researchers further suggest that supplementation in the absence of deficiency may be dangerous. For example, Ghezzi and colleagues (2017) propose that excess antioxidant supplements, that act by scavenging reactive oxygen species, might interfere with essential signalling pathways [30]. The role of micronutrient and antioxidant supplementation as general adjunctive therapy for patient populations with known micronutrient deficiency risk, such as HIV patients, remains widely debated [7].

Baseline micronutrient deficiencies were common among participants of this study. Approximately 23.1% of participants had serum carotene levels <1mmol/L, 14.9% had deficient folate levels, and 3.7% had low levels of vitamin $B_{12}$. A large proportion of study participants had insufficient (<75nmol/L, 68.2%) or deficient (<40nmol/L, 21.4%) baseline vitamin D levels, although average values observed were comparable to the general adult population [31]. Overall, prevalence of low micronutrient levels diminished with supplementation over the trial period, but no significant difference was found between the high-dose and RDA supplement groups, perhaps suggesting no added benefit of high-dose supplementation in reducing the frequency of low levels of some micronutrients.

A recent meta-analysis of 8 studies reported that HIV-positive individuals that experience inadequate nutrition tended to have an average of 91 fewer CD4 T lymphocytes/μl compared to their food-secure counterparts [32]. Within our patient population, median baseline CD4 T lymphocyte count was 495 cells/μl (IQR:429–610). We determined that the rate of CD4 T lymphocyte decline and time from randomization to ART initiation did not differ between groups. The rate of CD4 T lymphocyte decline appeared slower in the Treatment group, however this did not reach statistical significance, and was insignificant in a per-protocol sensitivity analysis. These results contrast findings by Baum and colleagues (2013) that showed that ART-naïve patients in Botswana receiving high-dose multivitamin plus selenium supplementation had a lower risk of reaching a CD4 T lymphocyte count at or below 250 cells/μL or AIDS-defining illness or death compared to placebo [11]. An RCT in Tanzania found that the absolute risk of HIV-disease progression or death was the same (72%) for HIV-positive patients on ART receiving standard doses of micronutrient supplementation and those receiving high-dose supplementation [33]. Interestingly, this group also found that high-dose supplementation had no effect on CD4 T lymphocyte count or plasma viremia, but increased the risk of elevated liver enzymes. That study was stopped prematurely due to significant elevation of serum ALT in the high-dose treatment group [33]. In the present study, both groups had positive mean changes in ALT levels over time (Control: 3.037 IU/L/52 weeks; Treatment: 0.936 IU/L/52 weeks); and there was no statistical significance between the difference in mean slope for the

Control versus Treatment group (p = 0.6325) (S7 Fig). Of note, the Isanaka study differs from the current study, since they looked at HIV-infected individuals that were initiating ART at the beginning of the RTC [33].

Although this study has not shown significant benefits to a high-dose supplement compared to an RDA supplement, we do not diminish the importance of adequate nutrition in the maintenance of good health in the context of HIV infection. Specific micronutrient replacement or supplementation should be practiced for adult HIV patients in whom a deficiency has been identified. Although not measured in this study, a potential benefit of high-dose micronutrient supplementation has been shown to decreased incidence of peripheral neuropathy, a known sequela of HIV infection and/or ART, compared to those receiving standard dose supplementation [33]. Some other supplements, for instance high-dose vitamin D3 (with calcium) has been shown to mitigate the negative impact of ART initiation on bone (mineral) density in a study among persons living with HIV using efavirenz/emtricabine/tenofovir disoproxil fumarate [34], which was not examined in our study. Yet, our data suggest no advantage and of high-dose supplement in ART-naïve HIV patients.

## Limitations

The strengths of this trial include rigorous design, conduct and analysis; high levels of study supplement adherence; demonstrated supplement stability; and adequate statistical power in design to detect a clinically significant effect, if there was one to be had. However, this study has limitations that must be taken into consideration. The Control group took an RDA multivitamin, not a placebo, and comparative results must be interpreted with that in mind. The use of a standard treatment control group allowed us to avoid the ethical concerns of not providing treatment, as in a placebo-controlled trial, but without having a concurrent placebo arm to compare with, we relied on indirect evidence on standard therapy effect.

Although study supplement adherence was assessed indirectly by several measures, its metrics remain subject to imprecision and biases. However, this trial suffers less risk of bias than the body of evidence thus far. An interim analysis revealed that there was no reasonable prospect of detecting a significant difference in outcome with the intended sample size, thus recruitment to the trial was stopped prematurely for futility even though follow-up was completed. The majority of our study population were middle-aged White males across Canada, limiting generalizability to other populations and contexts. Although high-dose micronutrient and antioxidant treatments are still widely used, the standard treatment recommendations for HIV no longer include ART deferment until a threshold CD4 T lymphocyte count decline has been observed [35], the context in which this trial was initiated.

## Conclusion

This study has identified no significant benefit or harm of high-dose micronutrient, mineral, trace element and antioxidant supplementation on HIV-associated immune deficiency progression, when compared to standard RDA micronutrient supplementation in untreated, HIV-positive patients.

## Supporting information

**S1 Checklist. CONSORT 2010 checklist of information to include when reporting a randomised trial**\*.
(DOC)

**S1 Fig. Kaplan-Meier Curve: Time to \*confirmed (in 2 measures) CD4 T lymphocyte decline < 350 cells/μL with sensitivity analysis censoring for individuals off-protocol.** Week 0 includes all individuals who were screened in and allocated to a group. The table beneath shows the individuals remaining at risk at each time point and the number of individuals experiencing an event in that interval is in brackets. There were 11 events in the Control group (Black line) and 16 events in the Treatment group (Gray line) over the study period of 96 weeks.
(PPTX)

**S2 Fig. Kaplan-Meier Curve: Time to ART start with sensitivity analysis censoring for individuals off-protocol.** Week 0 includes all individuals who were screened in and allocated to a group. The table beneath shows the individuals remaining at risk at each time point and the number of individuals experiencing an event in that interval is in brackets. There were 29 events in the Control group (Black line) and 28 events in the Treatment group (Gray line) over the study period of 96 weeks.
(PPTX)

**S3 Fig. Kaplan-Meier Curve: Time to any primary outcome event (CD4<350, ART, AIDS or death) with sensitivity analysis censoring for individuals off-protocol.** Week 0 includes all individuals who were screened in and allocated to a group. The table beneath shows the individuals remaining at risk at each time point and the number of individuals experiencing an event in that interval is in brackets. There were 32 events in the Control group (Black line) and 34 events in the Treatment group (Gray line) over the study period of 96 weeks.
(PPTX)

**S4 Fig.** CD4 T lymphocyte counts of HIV-infected participants on 100% RDA (Control- A, C) versus high-dose (Treatment- B, D) supplements over time, confined to intent-to-treat analysis (data censored for those starting ART, ITT; A, B) or off-protocol censoring (OP; C, D). For graphs A-D, each line represents an individual participant's measurements taken every 12 weeks until the end of the study (96 weeks or discontinuation of study) and only data from those participants with measurements for at least 36 weeks (i.e. having at least 3 data points) were included. Measurements at Week 0 represent the participant's baseline CD4 T lymphocyte count prior to taking the indicated supplement. E) Mean change in CD4 T lymphocytes (cells/μL) over time (in Weeks) was calculated for Control and Treatment groups using linear mixed-effects model analysis. Data was censored for ITT or OP and the rate per 52 weeks is given. F) The mean difference in slope at 52 weeks for Control versus Treatment CD4 trajectories with the 95% confidence intervals are reported. Both ITT and OP censoring are reported with p-values.
(PPTX)

**S5 Fig. CD4 T lymphocyte count, CD8 T lymphocyte count (A) and CD4:CD8 ratios (B) analyzed using linear mixed-effects model analysis were not significantly different between Control and Treatment groups.** The mean slope for CD4, CD8 and the CD4:CD8 ratio were calculated for Control and Treatment groups (all data was censored for those off-protocol (OP)). A) The difference in the mean slopes for Control versus Treatment are presented for CD4 and CD8 T lymphocytes over 52 weeks with 95% confidence interval (capped bars) and p value. B) The difference in the mean slopes for the CD4:CD8 ratio over 52 weeks is reported with 95% confidence interval (capped bars) and p value.
(PPTX)

**S6 Fig.** HIV viral load of HIV-infected participants on 100% RDA (Control- A, C) versus high-dose (Treatment-B, D) supplements over time, confined to intent-to-treat analysis (ITT,

data censored for those starting ART; A, B) or off-protocol censoring (OP; C, D). For graphs A-D, each line represents an individual participant's measurements taken every 12 weeks until the end of the study (96 weeks or discontinuation of study) and only data from those participants with measurements for at least 36 weeks (i.e. having at least 3 data points) were included. Measurements at Week 0 represent the participant's baseline HIV viral load prior to taking the indicated supplement. E) Linear mixed-effect model analysis was used to calculate the mean change in HIV viral load (Log10 copies/mL) over time (in Weeks) for Control and Treatment groups. Data was censored for ITT or OP and the rate per 52 weeks is given. F) The difference in mean slope for HIV viral load trajectories of Control versus Treatment groups are reported with the 95% confidence interval as the capped bars and p-values listed beside.
(PPTX)

**S7 Fig. Serum chemistries are not significantly different between Control and Treatment groups (off-protocol censoring, OP).** A) Mean slopes for each serum chemistry were calculated for Control and Treatment groups using linear mixed-effects model analysis. B) The difference in mean slope for serum chemistry trajectories of Control versus Treatment groups are reported in change in units (given for each serum chemistry) per 52 weeks, with the 95% confidence interval being reported as the capped bars and p values listed beside.
(PPTX)

**S8 Fig. Self-measured Quality of Life (QOL) assessment for participants taking 100% RDA (Control) versus high-dose (Treatment) supplements over time, confined to per-protocol analysis.** Participants were asked to grade their QOL at baseline and each study visit (every 12 weeks up to 96 weeks) using the Euro-QoL 5 dimensions—3 level questionnaire. The parameters of activity, anxiety, mobility, pain and self-care were graded as no problem (1), some problems (2) or extreme problems (3). The bars represent the percentage of individuals with no problems in the Control (black bars) versus Treatment (gray bars) groups for (A) Activity, (B) Anxiety, (C) Mobility, (D) Pain, and (E) Self-Care (out of 100% total). Data was censored for participants off-protocol. The n values for Control and Treatment groups at each time point are listed in the table below the respective graph. No statistics were calculated, as these are a change in proportion rather than individual changes over time.
(PPTX)

**S9 Fig. The Complete MAINTAIN Study Team.**
(PPTX)

**S1 Table. Supplement Stability Report per 16 capsules (actual daily dose) over the course of the study.** Treatment refers to the High-dose supplement and Control refers to the 100% recommended daily allowance supplement. The readings are given in different units based on the micronutrient per 16 capsules (daily-dose given to participants).
(DOCX)

**S2 Table. Serious Adverse Events among MAINTAIN participants.** Note that the 29 serious adverse events happened to 19 individual participants. All the serious adverse events were reported to Health Canada and to the local REB. All were deemed unexpected and unrelated to the study treatment.
(DOCX)

**S3 Table. Albumin measurements (in blood) taken quarterly over the study period in control (100% recommended daily allowance supplement) and Treatment (High-dose supplement) groups.**
(DOCX)

**S4 Table. Alanine Transaminase (ALT) measurements (in blood) taken quarterly over the study period in Control (100% recommended daily allowance supplement) and Treatment (High-dose supplement) groups.**
(DOCX)

**S5 Table. Alkaline Phosphatase measurements (in blood) taken quarterly over the study period in control (100% recommended daily allowance supplement) and treatment (High-dose supplement) groups.**
(DOCX)

**S6 Table. Amylase measurements (in blood) taken quarterly over the study period in control (100% recommended daily allowance supplement) and Treatment (High-dose supplement) groups.**
(DOCX)

**S7 Table. Aspartate Transaminase (AST) measurements (in blood) taken quarterly over the study period in control (100% recommended daily allowance supplement) and Treatment (High-dose supplement) groups.**
(DOCX)

**S8 Table. Bilirubin (total) measurements (in blood) taken quarterly over the study period in control (100% recommended daily allowance supplement) and treatment (High-dose supplement) groups.**
(DOCX)

**S9 Table. Blood Glucose (random) measurements taken quarterly over the study period in control (100% recommended daily allowance supplement) and treatment (High-dose supplement) groups.**
(DOCX)

**S10 Table. Blood urea nitrogen (BUN) measurements taken quarterly over the study period in control (100% recommended daily allowance supplement) and treatment (High-dose supplement) groups.**
(DOCX)

**S11 Table. C-reactive protein (CRP) measurements (in blood) taken quarterly over the study period in control (100% recommended daily allowance supplement) and treatment (High-dose supplement) groups.**
(DOCX)

**S12 Table. Creatinine measurements (in blood) taken quarterly over the study period in control (100% recommended daily allowance supplement) and treatment (High-dose supplement) groups.**
(DOCX)

**S13 Table. Total Protein measurements (in blood) taken quarterly over the study period in control (100% recommended daily allowance supplement) and Treatment (High-dose supplement) groups.**
(DOCX)

**S1 File. Revised without Logos—OHREB-Application.07Dec2007.**
(DOC)

**S2 File. Supplementary Methods_June10th_2020.**
(DOCX)

**S3 File.**
(XLS)

## Acknowledgments

The development of the study protocol was supported by Board Directed Funds from the Ontario HIV Treatment Network (OHTN). A Community Advisory Committee, comprising Ron Rosenes (chair), Beverly Deutsch and Devan Nambiar, provided community perspective to the study protocol, the informed consent form and advertising material. The DSMB was comprised of Stephen Shafran (Chair), Phil Sestak, Sarah Rose, Hubert Wong, Joel Singer, and Clement Chabot. Thanks to Natasha Filoso-Timpson and Jennifer Molson for administrative assistance, and to Cheynne McLean and Rika Moorhouse. The DSMB was struck at the National Centre of the CIHR-CTN, had full access to interim data at interim analysis, and conducted planned, independent unblinded analyses at the request of the PI (DWC). Jonathan Angel was the medical advisor for the trial. Please see S11 Fig for the full MAINTAIN Study Team list.

WLW, LB, NS, DAF, HH, EJM, JS, DWC conceived and designed the MAINTAIN Trial; WLW, BC, MJG, DLPK, AR, EDR, RR, NT, DT, SLW, DWC conducted the research; JEM, RM, JS, KAM, NT, DV, performed the analyses; WLW, JEM, LB, BC, MJG, HH, KAM, RM, JS, DT, SLW, DWC drafted and revised the manuscript; DWC had primary responsibility for final content. All authors read and approved the final manuscript.

## Author Contributions

**Conceptualization:** Wendy L. Wobeser, Joel Singer, D. William Cameron.

**Data curation:** Wendy L. Wobeser, Louise Balfour, Brian Conway, M. John Gill, Harold Huff, Donald L. P. Kilby, Ranjeeta Mallick, Katherine A. Muldoon, Anita Rachlis, Edward D. Ralph, Darrell Tan, Nancy Tremblay, Dong Vo, D. William Cameron.

**Formal analysis:** Wendy L. Wobeser, Louise Balfour, Brian Conway, M. John Gill, Donald L. P. Kilby, Dean A. Fergusson, Ranjeeta Mallick, Edward J. Mills, Katherine A. Muldoon, Neera Singhal, Sharon L. Walmsley, D. William Cameron.

**Funding acquisition:** D. William Cameron.

**Investigation:** Anita Rachlis, D. William Cameron.

**Methodology:** Dean A. Fergusson.

**Validation:** Ron Rosenes, Joel Singer, Darrell Tan.

**Visualization:** Joanne E. McBane, Neera Singhal.

**Writing – original draft:** Wendy L. Wobeser, Joanne E. McBane, Louise Balfour, M. John Gill, Edward J. Mills, Ron Rosenes, Darrell Tan, Sharon L. Walmsley, D. William Cameron.

**Writing – review & editing:** Joanne E. McBane, D. William Cameron.

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
