## [Decision Letter · Decision Letter 0]

25 Jan 2022

PONE-D-21-33096A randomized control trial of high-dose micronutrient-antioxidant supplementation in healthy persons with untreated HIV infectionPLOS ONE

Dear Dr. Cameron,

Thank you for submitting your manuscript to PLOS ONE. After careful consideration, we feel that it has merit but does not fully meet PLOS ONE’s publication criteria as it currently stands. Therefore, we invite you to submit a revised version of the manuscript that addresses the points raised during the review process.

We look forward to receiving your revised manuscript.

Kind regards,

Maret G Traber, PhD

Academic Editor

PLOS ONE

Journal Requirements:

Additional Editor Comments (if provided):

The expert reviewers have recommended some minor changes. Please address these.

Reviewers' comments:

Reviewer's Responses to Questions

**Comments to the Author**

1. Is the manuscript technically sound, and do the data support the conclusions?

Reviewer #1: Yes

Reviewer #2: Yes

2. Has the statistical analysis been performed appropriately and rigorously? 

Reviewer #1: Yes

Reviewer #2: Yes

3. Have the authors made all data underlying the findings in their manuscript fully available?

Reviewer #1: Yes

Reviewer #2: Yes

4. Is the manuscript presented in an intelligible fashion and written in standard English?

Reviewer #1: Yes

Reviewer #2: Yes

5. Review Comments to the Author

Reviewer #1: To some extent the findings are confirmatory in nature, but a large volume of work was presented which has been accumulated over a long period of time. The results carry sufficient merits to be published. Other questions are raised in my comments to authors.

Reviewer #2: Overall comments:

The design and methods appear to be described and performed to a high standard. Though I have some suggestions and a few questions below on a few places which I think need some clarification or more information.

Specific comments:

1. (line 21) Intent-to-treat censoring

2. (line 124) It wasn't clear to me what "PLOS ONE" means here. Is this a missing reference?

3. (lines 217) Please provide a citation for the mixed-effects model.

4. (line 218) Which correlation structure was used?

5. (lines 256-273) I think some or most of this section should be included in the methods. I'm not quite sure how much; my guess is the first two paragraphs (lines 256-269).

6. (lines 256-273) I would also encourage a more complete description of the methods used for the simulations that are described in this section. This may be more appropriate to be included in the supplementary methods.

7. (lines 315-318) It seems redundant to include the p-values for both the log-rank test and PH model.

8. (lines 217, 219, 222, 227, 320, etc.) At this point in the paper, I am finding myself confused with the use of the terms "censored" and "censoring". In statistics, censoring has a specific definition, meaning the data are not available or hidden. For continuous variables, a "censored" value usually is one that falls either below or above a device's detection ability. In HIV, this is typically associated with measurements of viral load where, at least historically, assays have had a lower limit of detection (LLOD). Participants with low viral loads, their values were "censored" or hidden because the measurement was below the LLOD. I'm guessing "censored" is being used for two different purposes here and I suggest the authors read through the paper and change one of the terms to make this clearer.

9. (Figures 2-4) Ideally, these figures would include confidence bands to depict the variability in these estimates. Also, I encourage you to include them in color so they are more striking and because there are no additional fees for color figures. If there is a worry about the figures showing up in grayscale, you could switch to a palette at https://colorbrewer2.org/.

6. PLOS authors have the option to publish the peer review history of their article (what does this mean?). If published, this will include your full peer review and any attached files.

Reviewer #1: **Yes: **I. Tong Mak

Reviewer #2: No

---

## [Author Response · Author response to Decision Letter 0]

11 May 2022

Review Comments to the Author

Specific comments:

1. (line 21) Intent-to-treat censoring

Thank you for noticing this typo – it has been corrected.

2. (line 124) It wasn't clear to me what "PLOS ONE" means here. Is this a missing reference?

The actual reference has been added to the manuscript.

3. (lines 217) Please provide a citation for the mixed-effects model.

We have added the required citation.

4. (line 218) Which correlation structure was used?

A variance component structure was used where each patient was considered independent across all patients and visit was nested within each patient where a different variance component was each visit – this was added to the manuscript.

5. (lines 256-273) I think some or most of this section should be included in the methods. I'm not quite sure how much; my guess is the first two paragraphs (lines 256-269).

Thank you for this suggestion, part of the section has been moved to methods.

6. (lines 256-273) I would also encourage a more complete description of the methods used for the simulations that are described in this section. This may be more appropriate to be included in the supplementary methods.

This has been added for clarification: “Both scenarios assumed outcome rates that served as the basis for the sample size calculation for the two groups which served as the basis for the original sample size calculation was true, and simulated outcomes for those participants on study and still at risk, as well as for participants yet to be recruited in the second scenario.” 

7. (lines 315-318) It seems redundant to include the p-values for both the log-rank test and PH model.

We have removed the redundant p value.

8. (lines 217, 219, 222, 227, 320, etc.) At this point in the paper, I am finding myself confused with the use of the terms "censored" and "censoring". In statistics, censoring has a specific definition, meaning the data are not available or hidden. For continuous variables, a "censored" value usually is one that falls either below or above a device's detection ability. In HIV, this is typically associated with measurements of viral load where, at least historically, assays have had a lower limit of detection (LLOD). Participants with low viral loads, their values were "censored" or hidden because the measurement was below the LLOD. I'm guessing "censored" is being used for two different purposes here and I suggest the authors read through the paper and change one of the terms to make this clearer.

Hope this will clarify the confussion – the data sets were “censored” and “censoring” was used in the manuscript to describe the process of excluding censored values. We tried to clarify this in the manuscript as well. 

9. (Figures 2-4) Ideally, these figures would include confidence bands to depict the variability in these estimates. Also, I encourage you to include them in color so they are more striking and because there are no additional fees for color figures. If there is a worry about the figures showing up in grayscale, you could switch to a palette at https://colorbrewer2.org/.

We are submitting versions in color this time.

REVIEWER #2

This manuscript summarized the results from a randomized, double-blind, placebo-controlled multicenter clinical trial in Canada to assess if high doses of micronutrient plus antioxidant supplementation might provide significant beneficial effects on the HIV disease progression in asymptomatic ART-naïve HIV patients compared to similar patients receiving RDA supplementation (control). Based primarily on time-to first detection of CD4 <350 cells/ul, or ART initiation or AIDS-defined illness, no significant difference in disease progression between these two groups were found. The study represents a large volume of work accumulated for over 2 years, and the trial protocol and analyses were considered rather rigorous and thus the data seem to be of high quality, and the findings were consistent with the majority opinions/reviews in the field. The treatment group also received high dose antioxidant supplementation which renders this study more unique and the results worthy noticing. Nevertheless, the impact on key indices of oxidative stress and antioxidant capacity are missing but desirable. Other issues are listed as following.

Specifics:

1. It is believed that the inclusion of multiple antioxidants and minerals in the treatment group (in addition to high doses of multivitamins) was designed to address the compelling hypothesis that oxidative stress plays an important role contributing to the pathogenesis of HIV/AIDS. However, key blood indices of oxidative stress (e.g., F2-isoprostane, 4HNE-His, TBARS etc., and/or antioxidant decreases (e.g., GSH, oxidant scavenging capacity etc.) are missing which were obtainable from the patients’ plasma/blood samples. It would be highly relevant to the theme of the study if giving RDA levels of multivitamins alone would be sufficient to improve the antioxidant/oxidative indices from the baseline comparing to giving high doses of multivitamins and antioxidants.

Thank you for this suggestion. As we saw no clinicaly relevant improvement in the course of the study we we did not pursue suggested analysis. Each site may have some of the data, but collecting this at this time would be impossible.

2. The rate of CD4 T cell decline appeared to be substantially slower in the treatment (-42.7 cells/ul vs -79.7 cells/ul per 52 weeks for the control); yet statistical significance was not reached due to the stringent protocol sensitivity analysis. In a published work, it was suggested that the beneficial effects of micronutrient supplementation could only be solidified if the HIV infection was at an earlier stage. It is unclear from the study described whether such a criterion (of early HIV infection) was taken into consideration for the patient selection. Or if the patients might come from different stages of HIV infection and thus contribute to a higher degree of variability for the CD 4 T cell decline.

All the participants had been diagnosed recently, with a mean diagnosis of within 2 years, however, how long the participant had been positive before diagnosis would have been n unknown variable. This clarification was added in the manuscript.

---

## [Editor Report · Decision Letter 1]

14 Jun 2022

A randomized control trial of high-dose micronutrient-antioxidant supplementation in healthy persons with untreated HIV infection

PONE-D-21-33096R1

Dear Dr. Cameron,

We’re pleased to inform you that your manuscript has been judged scientifically suitable for publication and will be formally accepted for publication once it meets all outstanding technical requirements.

Kind regards,

Maret G Traber, PhD

Academic Editor

PLOS ONE
---

## [Editor Report · Acceptance letter]

5 Jul 2022

PONE-D-21-33096R1 

A randomized control trial of high-dose micronutrient-antioxidant supplementation in healthy persons with untreated HIV infection. 

Dear Dr. Cameron:

I'm pleased to inform you that your manuscript has been deemed suitable for publication in PLOS ONE. Congratulations! Your manuscript is now with our production department. 

Kind regards, 

on behalf of

Professor Maret G Traber 

Academic Editor

PLOS ONE